# Enriching Complex Networks with Word Embeddings for Detecting Mild Cognitive Impairment from Speech Transcripts

## Abstract

Mild Cognitive Impairment (MCI) is a mental disorder difficult to diagnose. Although language impairment is an important marker it is frequently undervalued in cognitive assessments. Linguistic features, mainly from parsers, have been used to detect MCI. However, MCI disfluencies produce agrammatical speech impacting in parsing results; manually correcting transcripts of patient's speech is not a solution to large scale assessments. In this paper, we use complex network features to automatically identifying MCI in transcripts, using several classification algorithms, as it a lightweight and language independent representation. We modeled transcripts into complex networks and enriched them with word embeddings to better represent short texts produced in assessments. We evaluate our model in three datasets: one from the Dementia-Bank; Cinderella and Arizona-Battery in Portuguese were produced by assessments applied at University of São Paulo Medical School. The results show that complex networks are suitable to detect MCI and outperform linguistic features in all datasets. We also combined the classifiers through ensemble and multi-view learning. The ensemble of algorithms and multi-view method achieved 10% of improvement in Cinderella dataset compared to the best individual classifier. For the Arizona dataset, the multi-view achieved the highest accuracy (80%), a 4.71% of improvement.

## 1 Introduction

Mild Cognitive Impairment (MCI) can affect one or multiple cognitive domains (e.g. memory, language, visuospatial skills and the executive function), and may represent a pre-clinical stage of Alzheimer's disease (AD). The impairment that affects memory, referred to as amnestic MCI, is the most frequent, with the highest conversion rate for AD, at 15% a year versus 1 to 2% for the general population. Since dementias are chronic and progressive diseases, their early diagnosis ensures a greater chance of success to engage patients in non-pharmacological treatment strategies such as cognitive training, physical activity and socialization (Teixeira et al., 2012).

Language is one of the most efficient information sources for assessing cognitive functions. Changes in language usage are frequent in patients with dementia and are normally first recognized by the patients themselves or their family members. Therefore, the automatic analysis of discourse production is promising for diagnosing MCI at early stages, which may address potentially reversible factors (Muangpaisan et al., 2012). Proposals to detect language-related impairment in dementias include machine learning (Jarrold et al., 2010; Roark et al., 2011; Fraser et al., 2014, 2015), magnetic resonance imaging (Dyrba et al., 2015), and data screening tests added to demographic information (Weakley et al., 2015). The discourse production (mainly narratives) is attractive because it allows for the analysis of linguistic microstructures, including phonetic-phonological, morphosyntactic and semantic-lexical components, as well as semantic-pragmatic macrostructures.

Automated discourse analysis based on Natural Language Processing (NLP) resources and tools to diagnose dementias via machine learning meth-

ods have been used for the English language (Lehr et al., 2012; Jarrold et al., 2014; Orimaye et al., 2014; Fraser et al., 2015; Davy et al., 2016) and for Brazilian Portuguese (Aluísio et al., 2016). A variety of features are required for this analysis, including Part-of-Speech (PoS), syntactic complexity, lexical diversity and acoustics features. Producing robust tools to extract these features is extremely difficult because speech transcripts used in neuropsychological evaluations contain disfluencies (repetitions, revisions, paraphasias) and patient's comments about the task being evaluated. Another problem in using linguistic knowledge is the high dependence on manually created resources, such as hand-crafted linguistic rules and/or annotated corpora. Even when traditional statistical techniques (Bag of Words (BoW) or ngrams) are applied they also have problems to deal with disfluencies.

An approach applied successfully to several areas of NLP (Mihalcea and Radev, 2011), which may suffer less from the problems mentioned above, relies on the use of complex networks and graph theory. The word adjacency network model (i Cancho and Solé, 2001; Roxas and Tapang, 2010; Amancio et al., 2012a; Amancio, 2015b) has provided good results in text classification (de Arruda et al., 2016) and related tasks, namely author detection (Amancio, 2015a), identification of literary movements (Amancio et al., 2012c), authenticity verification (Amancio et al., 2013) and sense discrimination (Amancio et al., 2012b).

In this paper, we show that speech transcripts (narratives or descriptions) can be modeled into complex networks that are enriched with word embeddings. We modeled narratives and short descriptions into complex networks and enriched them with word embeddings to better represent short texts produced in these assessments. When applied to a machine learning classifier, the complex network features were able to distinguish between control participants and mild cognitive impairment participants. Discrimination of the two classes could be improved by combining complex networks with linguistic and traditional statistical features; we also identified the best scenario for application of network features extracted from transcripts of neuropsychological assessments.

## 2 Related Work

Detection of memory impairment has been based on linguistic, acoustic, and demographic features, in addition to scores of neuropsychological tests. Linguistic and acoustic features were used to automatically detect aphasia (Fraser et al., 2014); and AD (Fraser et al., 2015) or dementia (Orimaye et al., 2014) in public corpora of Dementia-Bank. Other studies distinguished differents types of dementia (Garrard et al., 2014; Jarrold et al., 2014), in which speech samples were elicited using the Picnic picture of the Western Aphasia Battery. Davy et al. (2016) also used the Picnic scene for detecting MCI, where the subjects were asked to write (by hand) a detailed description of the scene.

As for automatic detection of MCI in narrative speech, Roark et al. (2011) extracted speech features and linguistic complexity measures of speech samples obtained with the Wechsler Logical Memory (WLM) subtest (Wechsler et al., 1997), and Lehr et al. (2012) fully automatized the WLM subtest. Some studies used short animated films to evaluate immediate and delayed recall in MCI patients which were asked to talk about the first film shown, then about their previous day, and finally about another film shown last. Tóth et al. (2015) adopted automatic speech recognition (ASR) to extract a phonetic level segmentation, which they used to calculate acoustic features. Vincze et al. (2016) used speech, morphological, semantic, and demographic features collected from their speech transcripts to automatically identify patients suffering from MCI.

For the Portuguese language, machine learning algorithms were used to identify subjects with AD and MCI. (Aluísio et al., 2016) used a variety of linguistic metrics, such as syntactic complexity, idea density (da Cunha et al., 2015), and text cohesion through latent semantics. PLN tools with high precision are needed to compute these metrics, which is a problem for Portuguese since there is no robust dependency or constituency parser. Therefore, the transcriptions had to be manually revised; they were segmented in sentences, following a semantic-structural criterion and capitalization was added afterwards. The authors also removed disfluencies and inserted omitted subjects when they were hidden, in order to reduce parsing errors. This process is obviously expensive, which has motivated us to use complex networks in the

present study to model transcriptions and avoid a manual preprocessing step.

## 3 Modeling and Characterizing Texts as Complex Networks

The theory and concepts of complex networks have been used in several NLP tasks (Mihalcea and Radev, 2011; Cong and Liu, 2014), such as text classification (de Arruda et al., 2016), summarization (Antiqueira et al., 2009; Amancio et al., 2012a) and word sense disambiguation (Silva and Amancio, 2012). In this study, we used the word co-occurrence model (also called the word adjacency model) with a small modification because most of the syntactical relations occur among neighbouring words (i Cancho et al., 2004). Each distinct word becomes a node and words that are adjacent in the text are connected by an edge. Mathematically, a network is defined as an undirected graph $G = \{V, E\}$, formed by a set $V = \{v_1, v_2, ..., v_n\}$ of nodes (words) and a set $E = \{e_1, e_2, ..., e_m\}$ of edges (co-occurrence) that are represented by an adjacency matrix $A$, whose elements $A_{ij}$ are equal to 1 whenever there is an edge connecting nodes (words) $i$ and $j$, and equal to 0 otherwise.

Before modeling texts into complex networks, it is often necessary to do some preprocessing in the raw text. Preprocessing starts with tokenization where each document/text is divided into tokens (meaningful elements, e.g.: words and punctuation marks) and then *stopwords* and punctuation marks are removed, since they have little semantic meaning. One last step we decided to eliminate from the preprocessing pipeline is lemmatization, which transforms each word into its canonical form. This decision was made based on two factors: first, recent work has shown that lemmatization has little or no influence when network modeling is adopted in related tasks (Machicao et al., 2016). Second, the lemmatization process requires part-of-speech (POS) tagging that may introduce undesirable noises/errors in the text, since the texts in our work are transcriptions containing disfluencies.

Another problem with transcriptions in our work is their size. As demonstrated by Amancio (2015c), the classification of small texts using networks can be impaired, since short texts have almost linear networks, and the topological measures of these networks have little or no informa-

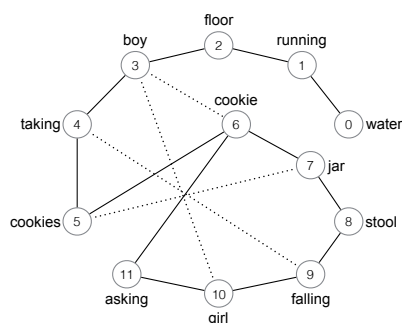

Figure 1: Example of co-occurrence network enriched with semantic information for the following transcription: "*The water's running on the floor. Boy's taking cookies out of cookie out of the cookie jar. The stool is falling over. The girl was asking for a cookie.*". The solid edges of the network represent co-occurrence edges and the dotted edges represent connections between words that had similarity higher than 0.5.

tion relevant to classification. To solve this problem, we adapted the approach of modeling networks with word embeddings proposed by Perozzi et al. (2014) to enrich the networks with semantic information. In this process of word embedding, language networks are generated from continuous word representations, where each word is represented by a dense, real-valued vector obtained by trainning neural networks in the language model task (or variations, such as context prediction) (Bengio et al., 2003; Collobert et al., 2011; Mikolov et al., 2013a,b). This structure is known to capture syntatic and semantic information (Mikolov et al., 2013a,b). Perozzi et al. (2014), in particular, takes advantage of word embeddings to build networks where each word is a vertice and edges are defined by similarity between words established by the proximity of the word vectors.

Following this methodology, in our model we added new edges to the co-occurrence networks considering similarities between words, that is, for all pairs of words in the text that were unconnected, an edge was created if their vectors (from word embedding) had a cosine similarity higher than a given threshold. Figure 1 shows an example of a co-occurrence network enriched by similarity links (the dotted edges). The gain in information by enriching a co-occurrence network with seman-

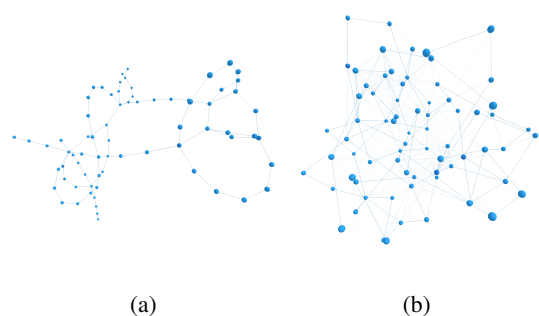

(a)                              (b)

Figure 2: Example of (a) co-occurrence network created for a transcript of the Cookie Theft dataset (see Supplementary Information, Section A1) and (b) the same co-occurrence network enriched with semantic information. Note that (b) is a more informative network than (a), since (a) is practically a linear network.

tic information is readily apparent in Figure 2.

## 4 Datasets, Features and Methods

### 4.1 Datasets

The datasets consisted of: (i) transcribed samples, manually segmented of the DementiaBank[1] and Cinderella story and (ii) transcribed samples of ABCD battery which were automatically segmented, since we are working towards a fully automated system to detect MCI in transcripts. The first dataset is composed of short English descriptions, while the second contains longer Brazilian Portuguese narratives. A third dataset had very short narratives, also in Portuguese. Below, we describe in more detail the datasets, participants, and the task in which they were used.

### 4.1.1 The Cookie Theft Picture Description Dataset

The clinical dataset used for English was created during a longitudinal study conducted by the University of Pittsburgh School of Medicine on Alzheimer's and related dementia, funded by the National Institute of Aging. To be eligible for inclusion in the study, all participants were required to be above 44 years of age, have at least 7 years of education, no history of nervous system disorders or be taking neuroleptic medication, have an initial Mini-Mental State Exam (MMSE) score of 10 or greater, and be able to give informed consent. The dataset contains transcripts of verbal interviews with AD and related Dementia patients,

---

[1] https://talkbank.org/DementiaBank/

including those with MCI (for more detail, see (Becker et al., 1994)).

We used 43 transcripts of patients with MCI, sampled from 326 people diagnosed with probable AD; 21 with possible AD. Table 1 shows the demographic information for the two diagnostic groups.

| Demographic | Control | MCI |
|---|---|---|
| Avg. Age (SD) | 64.1 (7.2) | 69.3 (8.2) |
| No. of Male/Female | 23/20 | 27/16 |

Table 1: Demographic information of participants in the Cookie Theft dataset and statistics of the dataset.

For this dataset, interviews were conducted in English and narrative speech was elicited using the Cookie Theft picture (Goodglass et al., 2001) (Figure 3 from Goodglass et al. (2001) in Section A.1). During the interview, patients were given the picture and were told to discuss everything they could see happening in the picture. The patients' verbal utterances were recorded and then transcribed into the CHAT (Codes for the Human Analysis of Transcripts) transcription format (MacWhinney, 2000).

We extracted the word-level transcript patient sentences from the CHAT files and discarded the annotation, as our goal was to create a fully automated system that does not require the input of a human annotator. We automatically removed filled pauses such as *uh*, *um* , *er* , and *ah* (e.g. *uh it seems to be summer out*), short false starts (e.g. *just t the ones* ), and repetition (e.g. *mother's finished certain of the the dishes* ), like (Fraser et al., 2015). The Control group has an average of 9.58 sentences per narrative, with each sentence having an average of 9.18 words; as for the MCI group, it has an average of 10.97 sentences per narrative, with 10.33 words per sentence in average.

### 4.1.2 The Cinderella Narrative Dataset

The dataset examined in this study included 20 subjects with MCI and 20 subjects without MCI for control, as diagnosed at the Medical School of the University of São Paulo (FMUSP). Table 2 shows the demographic information of the two diagnostic groups, which were also used in (Aluísio et al., 2016).

The criteria used to diagnose MCI came from (Petersen, 2004). Diagnostics were done by a multidisciplinary team with psychiatrists, geria-

| Demographic | Control | MCI |
|---|---|---|
| Avg. Age (SD) | 74.8 (11.3) | 73.3 (5.9) |
| Avg. Years of Education (SD) | 11.4 (2.6) | 10.8 (4.5) |
| No. of Male/Female | 27/16 | 29/14 |

Table 2: Demographic information of participants in the Cinderella dataset. The Avg. Education is given in years.

tricians, neurologists, neuropsychologists, speech pathologists, and occupational therapists, by a criterion of consensus. Inclusion criteria for the control group were elderly people with no cognitive deficits and preservation of functional capacity in everyday life. The exclusion criteria for the normal group were: poorly controlled clinical diseases, sensitive deficits that are not being compensated for and interfere with the performance in tests, other neurological or psychiatric diagnoses that are associated with dementia or cognitive deficits and use of medications in doses that affect cognition.

Speech narrative samples were elicited by having participants tell the Cinderella story; participants were given as much time as they needed to examine a picture book illustrating the story (Figure 4 in Section A). When each participant had finished looking at the pictures, the examiner asked the subject to tell the story in their own words, as in (Saffran et al., 1989). The time was recorded, but there was no limit imposed to the narrative length. If the participant had difficulty initiating, continuing speech, or presenting a long pause, an evaluator used a stimulus question "What happens next ?", seeking to encourage the participant to continue his/her narrative. When the subject was unable to proceed with the narrative, the examiner asked if he/she had finalized the story and had something to add. Each speech sample was recorded and then manually transcribed at the word level following the NURC/SP N. 338 EF and 331 D2 transcription norms.

Other tests were applied after the narrative in the following sequence: phonemic verbal fluency test, action verbal fluency, Camel and Cactus test (Bozeat et al., 2000), and Boston Naming test (Kaplan et al., 2001), in order to diagnose the groups.

Since our goal was to create a fully automated system that did not require the input of a human annotator, we segmented sentences to simulate a high-quality ASR transcript with sentence seg-

mentation, and we automatically removed the disfluencies following the same guidelines of Talk-Bank project[2]. However, other disfluencies (revisions, elaboration, paraphasias and comments about the task) were kept. The Control group has an average of 30.80 sentences per narrative, and each sentence averages 12.17 words; as for the MCI group, it has an average of 29.90 sentences per narrative, and each sentence averages 13.03 words.

We also evaluated a different version of this dataset used in Aluísio et al. (2016), where narratives were manually revised to improve parsing results: agramatical sentences and all the patients' comments not related to the story were removed, and omitted subjects were inserted. In this dataset, the Control group has an average of 45.10 sentences per narrative, and each sentence averages 8.17 words. The MCI group has an average of 31.40 sentences per narrative, with each sentence averaging 10.91 words.

### 4.1.3 The Arizona Battery for Communication Disorders of Dementia (ABCD) Dataset

The ABCD dataset examined included 23 subjects with MCI and 20 subjects without MCI for control, as diagnosed at the Medical School of the University of São Paulo (FMUSP). MCI subjects produced 46 narratives and the controls 39 ones. We used the automatic sentence segmentation method referred to as DeepBond (Treviso et al., 2017). Table 3 shows the demographic information. The Control group has an average of 5.23 sentences per narrative, with 11 words per sentence in average, and the MCI group has an average of 4.95 sentences per narrative, with an average of 12.04 words per sentence. Interviews were conducted in Portuguese and the subject listened to the examiner read a brief narrative. The subject then retold the narrative to the examiner twice: once immediately upon hearing it and again after a 30-minute delay (Bayles and Tomoeda, 1991). Each speech sample was recorded and then manually transcribed at the word level following the NURC/SP N. 338 EF and 331 D2 transcription norms.

### 4.2 Features

Features of three distinct natures were used for classifying the transcribed texts: topological met-

---

[2]https://talkbank.org/

| Demographic | Control | MCI |
|---|---|---|
| Avg. Age (SD) | 61 (7.5) | 72,0 (7.4) |
| Avg. Years of Education (SD) | 16 (7.6) | 13.3 (4.2) |
| No. of Male/Female | 6/14 | 16/7 |

Table 3: Demographic information of participants in the ABCD dataset. The Avg. Education is given in years.

rics of co-occurrence networks, linguistic features and statistics of bag of words representation.

### 4.2.1 Topological Characterization of Networks

Each transcription was mapped into a co-occurrence network, and then enriched via word embedding using the cosine similarity of words. Since the occurrence of out-of-vocabulary words is common in texts, we used the method proposed by (Bojanowski et al., 2016) to generate word embeddings. This method extends the skip-gram model to use character-level information, with each word being represented as a bag of character $n$-grams. It provides some improvement in comparison with the traditional skip-gram model in syntactic evaluation (Mikolov et al., 2013b) but not for the semantic evaluation.

Once the network has been enriched with semantic information, we characterize its topology using the following ten measurements:

1. **PageRank:** is a centrality measurement that reflects the relevance of a node based on its connections to other relevant nodes (Brin and Page, 1998);

2. **Betweenness:** is a centrality measurement that considers a node as relevant if it is highly accessed via shortest paths. The betweenness of a node $v$ is defined as the fraction of shortest paths going through node $v$;

3. **Eccentricity:** of a node is calculated by measuring the shortest distance from the node to all other vertices in the graph and taking the maximum;

4. **Eigenvector centrality:** is a measurement that defines the importance of a node based on its connectivity to high-rank nodes;

5. **Average Degree of the Neighbors of a Node:** is the average of the degrees of all its direct neighbors;

6. **Average Shortest Path Length of a Node:** is the average distance between this node and all other nodes of the network;

7. **Degree:** is the number of edges connected to the node;

8. **Assortativity Degree:** or degree correlation measures the tendency of nodes to connect to other nodes that have similar degree;

9. **Diameter:** is defined as the maximum shortest path;

10. **Clustering Coefficient:** measures the probability that two neighbors of a node are connected.

Most of the measurements described above are local measurements, i.e. each node $i$ possesses a value $X_i$, so we calculate the average $\mu(X)$, standard deviation $\sigma(X)$ and skewness $\gamma(X)$ for each measurement (Amancio, 2015b).

### 4.2.2 Linguistic Features

Linguistic features for classification of neuropsychological assessments have been utilized in several works (Roark et al., 2011; Jarrold et al., 2014; Fraser et al., 2014; Orimaye et al., 2014; Fraser et al., 2015; Vincze et al., 2016; Davy et al., 2016). We used the Coh-Metrix[3](Graesser et al., 2004) tool to extract features of English transcripts, resulting in 106 features. The metrics are divided into eleven categories: Descriptive, Text Easability Principal Component, Referential Cohesion, Latent Semantic Analysis (LSA), Lexical Diversity, Connectives, Situation Model, Syntactic Complexity, Syntactic Pattern Density, Word Information, Readability (Flesch Reading Ease, Flesch-Kincaid Grade Level, Coh-Metrix L2 Readability).

For Portuguese, Coh-Metrix-Dementia (Aluísio et al., 2016) was used. The metrics affected by constituency and dependency parsing were not used because they are not robust to deal with disfluencies. Metrics based on manual annotation (such as proportion short pauses, mean pause duration, mean number of empty words, and others) were also discarded. The metrics of Coh-Metrix-Dementia are divided into twelve categories: Ambiguity, Anaphoras, Basic Counts, Connectives,

---

[3] cohmetrix.com

Co-reference Measures, Content Word Frequencies, Hypernyms, Logic Operators, Latent Semantic Analysis, Semantic Density, Syntactical Complexity, and Tokens. The metrics used are shown in detail in Section A.2. In total, 58 metrics were used, from the 73 available in the website[4].

### 4.2.3 Bag of Words

The representation of text collections under the Bag-of-Words (BoW) assumption (i.e., with no information relating to word order) has been a robust solution for text classification by capturing word content- and frequency-specific differences that are relevant to the categories under investigation (Joachims, 1998; Drucker et al., 1999). In this methodology, transcripts are represented by a table in which the columns represent the terms (or existing words) in the transcripts and the values represent frequency of a term in a document, such as binary weights, term frequency (tf) or term frequency – inverse document frequency (tf–idf) (Salton, 1989). In this work term frequency was used.

### 4.3 Classification Algorithms

In order to quantify the ability of the topological characterization of networks, linguistic metrics and BoW features to distinguish subjects with MCI from those without, we employed four machine learning algorithms to induce classifiers from a training set. The techniques are Gaussian Naive Bayes (GaussianNB), k-Nearest Neighbor (K-NN), Support Vector Machine (SVM), linear and radial bases functions (RBF), and Random Forest (RF). We also combine these classifiers through ensemble and multi-view learning. In ensemble learning, multiple models/classifiers are generated and combined using, e.g. majority vote or average of class probabilities, to produce a single result (Zhou, 2012). In multi-view learning, multiple classifiers are trained in different feature spaces and thus combined to produce a single result. This approach is an elegant solution rather than combining all features in the same vector or space, primarily because the combination is not a straightforward step, which can lead to noise insertion, since the data have different natures. Secondly, using different classifiers for each feature space allows different weights to be given for each type of feature, weights that can be learned by a

---

regression method to improve the model. This approach has been successful for NLP (Collobert et al., 2011), sentiment classification (Xia et al., 2011) and pedestrian detection in images (Oliveira et al., 2010; Kim et al., 2016).

## 5 Experiments and Results

All experiments were conducted using the Scikit-learn[5] (Pedregosa et al., 2011), with classifiers evaluated on the basis of classification accuracy or the total proportion of narratives which were correctly classified. The evaluation was performed using 5-fold cross validation, and the threshold parameter was optimized with best values being $0.7$ in Cookie Theft dataset and $0.4$ in both Cinderella and ABCD dataset.

We trained the model proposed by Bojanowski et al. (2016) with default parametrs (100 dimensional embeddings, 5 size of the context window, 5 number of epochs). The accuracy in classification is given in Tables 4, 5, 6, and 7. CN, CNE, LM, and BoW denote, respectively, complex networks, complex network enriched with embeddings, linguistic metrics and Bag-of-Words, and CNE-LM, CNE-BoW, LM-BoW and CNE-LM-BoW refer to combinations of the feature spaces (multiview learning), using the majority vote. Cells with the – sign mean that it is not possible to apply majority voting. The last line represents the use of an ensemble of machine learning algorithms, in which the combination used was the majority voting in both ensemble and multiview learning.

The results for the three datasets show that characterizing transcriptions into complexity networks is competitive with other traditional methods, such as with the use of linguistic metrics. In fact, among the three types of features, using enriched networks (CNE) provided the highest accuracies, and in general CNE is better than using only complex networks. SVM gives better accuracy in most cases compared to other machine learning algorithms. As for ensemble and multi-view learnings there are some good results. For the Cookie Theft dataset, multi-view learning achieved the highest accuracy (65% of accuracy for narrative texts, a 2.3% of improvement compared to the best individual classifier). The ensemble of algorithms and multi-view CNE-LM achieved the highest accuracy in Cinderella dataset (75% of accuracy for narrative texts, a 10% of improvement compared

---

[4] http://143.107.183.175:22380

[5] http://scikit-learn.org

| Classifier | CN | CNE | LM | BoW | CNE-LM | CNE-BoW | LM-BoW | CNE-LM-BoW |
|---|---|---|---|---|---|---|---|---|
| SVM-Linear | 0.5250 | 0.5542 | 0.5694 | 0.5917 | – | – | – | 0.6056 |
| SVM-RBF | 0.5694 | **0.6264** | **0.5806** | **0.6014** | – | – | – | **0.6500** |
| KNN | **0.5903** | 0.6167 | 0.4611 | 0.5792 | – | – | – | 0.5903 |
| RF | 0.5222 | 0.4778 | 0.4569 | 0.4889 | – | – | – | 0.5000 |
| G-NB | 0.5125 | 0.4861 | 0.5694 | 0.5583 | – | – | – | 0.5000 |
| Ensemble | 0.5625 | 0.6042 | 0.5458 | 0.5806 | 0.5000 | 0.5625 | 0.5000 | **0.6500** |

Table 4: Classification accuracy achieved on Cookie Theft dataset.

| Classifier | CN | CNE | LM | BoW | CNE-LM | CNE-BoW | LM-BoW | CNE-LM-BoW |
|---|---|---|---|---|---|---|---|---|
| SVM-Linear | 0.5250 | 0.6000 | **0.5250** | 0.5000 | – | – | – | **0.5250** |
| SVM-RBF | **0.5750** | **0.6500** | 0.4750 | 0.3750 | – | – | – | 0.5000 |
| KNN | 0.4750 | 0.5000 | 0.4750 | 0.3750 | – | – | – | 0.3750 |
| RF | 0.5500 | 0.5750 | 0.4750 | 0.4500 | – | – | – | **0.5250** |
| G-NB | 0.4750 | 0.5250 | 0.4750 | **0.5500** | – | – | – | **0.5250** |
| Ensemble | 0.5250 | 0.6000 | 0.5000 | 0.3750 | **0.7500** | 0.5000 | 0.3750 | 0.4750 |

Table 5: Classification accuracy achieved on Cinderella dataset.

| Classifier | CNE | LM | BoW | CNE-BoW |
|---|---|---|---|---|
| SVM-Linear | 0.6500 | 0.6250 | 0.5250 | – |
| SVM-RBF | **0.6750** | **0.7250** | **0.5500** | – |
| KNN | 0.4750 | 0.5500 | 0.5000 | – |
| RF | 0.4750 | 0.5750 | 0.4500 | – |
| G-NB | 0.6500 | 0.6000 | 0.4500 | – |
| Ensemble | 0.6000 | 0.6750 | 0.4500 | **0.7500** |

Table 6: Classification accuracy achieved on Cinderella dataset manually processed to revise agrammatical sentences.

| Classifier | CNE | LM | BoW | CNE-LM-BoW |
|---|---|---|---|---|
| SVM-Linear | **0.7173** | 0.5904 | **0.7547** | **0.8018** |
| SVM-RBF | 0.6436 | 0.6819 | 0.5413 | 0.6583 |
| KNN | 0.6208 | **0.7056** | 0.5881 | 0.7171 |
| RF | 0.4921 | 0.6813 | 0.6265 | 0.6252 |
| G-NB | 0.5722 | 0.5744 | 0.6257 | 0.6450 |
| Ensemble | 0.6783 | 0.7056 | 0.6964 | 0.7636 |

Table 7: Classification accuracy achieved on ABCD dataset

to the best individual classifier). For the ABCD dataset, multi-view CNE-LM-BoW achieved the highest accuracy (80% of accuracy for narrative texts, a 4.71% of improvement compared to the best individual classifier). Somewhat surprising were the results of SVM with linear kernel in BoW feature space (75% of accuracy).

# 6 Conclusions and Future Work

In this study, we employed metrics of topological properties of CN in a machine learning classification approach to distinguish between healthy controls and patients with MCI. To the best of our knowledge, these metrics have never been used before to detect MCI in speech transcripts; CN were enriched with word embeddings to better represent short texts produced in neuropsychological assessments. We have shown that the topological properties of CN outperfom traditionally linguistic metrics, in individual classifiers results. Linguistic features depend on grammatical texts to present good results, as can be seen in the results of Cinderella dataset manually processed (Table 6). Furthermore, we found that combining machine and multi-view learning can improve accuracy. The accuracies found here are comparable to the values reported by other authors, ranging from 60% to 85% (Prud'hommeaux and Roark, 2011; Lehr et al., 2012; Tóth et al., 2015; Vincze et al., 2016), which means that it is not easy to distinguish between healthy subjects and those with cognitive impairments. The comparison with our results is not straightforward, though, because the databases used in the studies are different. There is a clear need of publicly available datasets to compare different methods, which would allow for optimizing detection of MCI in elderly people.

As future work, we intend to explore other methods to enrich CN, such as Recurrent Language Model language, and other metrics to characterize an adjacency network. The pursue of these strategies is relevant because language is one of the most efficient information sources for assessing cognitive functions, commonly used in neuropsychological assessments.

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

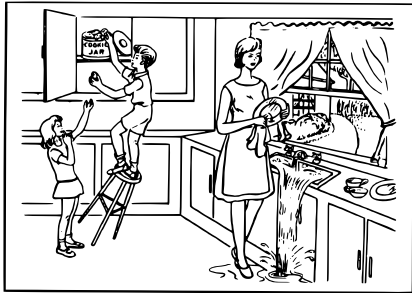

Figure 3: The Cookie Theft Picture, taken from the Boston Diagnostic Aphasia Examination (Goodglass et al., 2001).

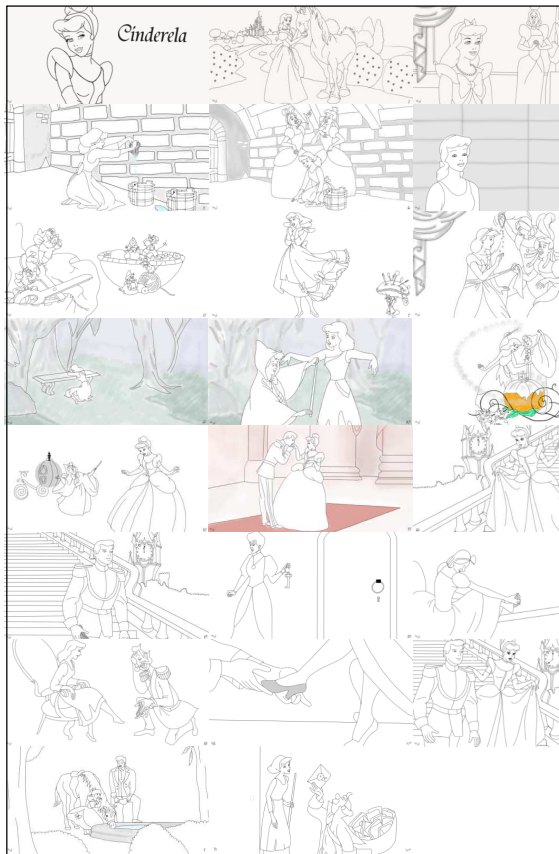

Figure 4: Cinderella Story Sequence of Pictures.

# A Supplementary Material

## A.1 Example of transcription

Below follows an example of a transcript of the Cookie Theft dataset.

You just want me to start talking ? Well the little girl is asking her brother we 'll say for a cookie . Now he 's getting the cookie one for him and one for her . He unbalances the step the little stool and he 's about to fall . And the lid 's off the cookie jar . And the mother is drying the dishes abstractly so she 's left the water running in the sink and it is spilling onto the floor . And there are two there 's look like two cups and a plate on the sink and board . And that boy 's wearing shorts and the little girl is in a short skirt . And the mother has an apron on . And she 's standing at the window . The window 's opened . It must be summer or spring . And the curtains are pulled back . And they have a nice walk around their house . And there 's this nice shrubbery it appears and grass . And there 's a big picture window in the background that has the drapes pulled off . There 's a not pulled off but pulled aside . And there 's a tree in the background . And the house with the kitchen has a lot of cupboard space under the sink board and under the cabinet from which the cookie you know cookies are being removed .

## A.2 Cohmetrix-Dementia metrics

1. **Ambiguity**: verb ambiguity, noun ambiguity, adjective ambiguity, adverb ambiguity;

2. **Anaphoras**: adjacent anaphoric references, anaphoric references;

3. **Basic Counts**: Flesch index, number of word, number of sentences, number of paragraphs, words per sentence, sentences per paragraph, syllables per content word, verb incidence, noun incidence, adjective incidence, adverb incidence, pronoun incidence, content word incidence, function word incidence;

4. **Connectives**: connectives incidence, additive positive connectives incidence, additive negative connectives incidence, temporal positive connectives incidence, temporal negative connectives incidence, casual positive connectives incidence, casual negative connectives incidence, logical positive connectives incidence, logical negative connectives incidence;

5. **Co-reference Measures**: adjacent argument overlap, argument overlap, adjacent stem overlap, stem overlap, adjacent content word overlap;

6. **Content Word Frequencies**: Content words frequency, minimum among content words frequency;

7. **Hypernyms**: Mean hypernyms per verb;

8. **Logic Operators**: Logic operators incidence, and incidence, or incidence, if incidence, negation incidence;

9. **Latent Semantic Analysis (LSA)**: Average and standard deviation similarity between pairs of adjacent sentences in the text, Average and standard deviation similarity between all sentence pairs in the text, Average and standard deviation similarity between pairs of adjacent paragraphs in the text, Givenness average and standard deviation of each sentence in the text;

10. **Semantic Density**: content density;

11. **Syntactical Complexity**: only cross entropy;

12. **Tokens**: personal pronouns incidence, type-token ratio, Brunet index, Honoré Statistics.

