# Peer review of "Enriching Complex Networks with Word Embeddings for Detecting Mild Cognitive Impairment from Speech Transcripts"

_ACL 2017 — decision unknown_

[Official Review · Reviewer 1 · rating 3 · confidence 4]
soundness 4 · originality 3 · clarity 3 · impact 4 · substance 3 · appropriateness 5 · meaningful comparison 3 · presentation format Oral Presentation

- Strengths:

This paper proposes to apply NLP to speech transcripts (narratives and
descriptions) in order to identify patients with MCI (mild cognitive
impairment, ICD-10 code F06.7). The authors claim that they were able to
distinguish between healthy control participants and patients with MCI (lines
141-144). However in the conclusion, lines 781-785, they say that “…
accuracy ranging from 60% to 85% …. means that it is not easy to distinguish
between healthy subjects and those with cognitive impairments”. So the paper
beginning is more optimistic than the conclusion but anyway the message is
encouraging and the reader becomes curious to see more details about what has
been actually done.

The corpus submitted in the dataset is constructed for 20 healthy patients and
20 control participants only (20+20), and it is non-understandable for people
who do not speak Portuguese. It would be good to incorporate more technological
details in the article and probably to include at least one example of a short
transcript that is translated to English, and eventually a (part of a) sample
network with embeddings for this transcript.

- Weaknesses:

The paper starts with a detailed introduction and review of relevant work. Some
of the cited references are more or less NLP background so they can be omitted
e.g. (Salton 1989) in section 4.2.3. Other references are not directly related
to the topic e.g. “sentiment classification” and “pedestrian detection in
images”, lines 652-654, and they can be omitted too. In general lines
608-621, section 4.2.3 can be shortened as well etc. etc. The suggestion is to
compress the first 5 pages, focusing the review strictly on the paper topic,
and consider the technological innovation in more detail, incl. samples of
English translations of the ABCD and/or Cindarela narratives.

The relatively short narratives in Portuguese esp. in ABCD dataset open the
question how the similarities between words have been found, in order to
construct word embeddings. In lines 272-289 the authors explain that they
generate word-level networks from continuous word representations. What is the
source for learning the continuous word representations; are these the datasets
ABCD+Cinderella only, or external corpora were used? In lines 513-525 it is
written that sub-word level (n-grams) networks were used to generate word
embeddings. Again, what is the source for the training? Are we sure that the
two kinds of networks together provide better accuracy? And what are the
“out-of-vocabulary words” (line 516), from where they come?

- General Discussion:

It is important to study how NLP can help to discover cognitive impairments;
from this perspective the paper is interesting. Another interesting aspect is
that it deals with NLP for Portuguese, and it is important to explain how one
computes embeddings for a language with relatively fewer resources (compared to
English). 

The text needs revision: shortening sections 1-3, compressing 4.1 and adding
more explanations about the experiments. Some clarification about the NURC/SP
N. 338 EF and 331 D2 transcription norms can be given.

Technical comments:

Line 029: ‘… as it a lightweight …’ -> shouldn’t this be ‘… as in
a lightweight …’

Line 188: PLN -> NLP

Line 264: ‘out of cookie out of the cookie’ – some words are repeated
twice 

Table 3, row 2, column 3: 72,0 -> 72.0

Lines 995-996: the DOI number is the same as the one at lines 1001-1002; the
link behind the title at lines 992-993 points to the next paper in the list

[Official Review · Reviewer 2 · rating 2 · confidence 5]
soundness 4 · originality 3 · clarity 2 · impact 2 · substance 2 · appropriateness 4 · meaningful comparison 3 · presentation format Poster

- Strengths:
This paper explores is problem of identifying patients with Mild Cognitive
Impairment (MCI) by analyzing speech transcripts available from three different
datasets. A graph based method leveraging co-occurrence information between
words found in transcripts is described. Features are encoded using different
characteristics of the graph lexical, syntactic properties, and many others. 
Results are reported using 5 fold cross validation using a number of
classifiers. Different models exhibit different performance across the three
datasets. This work targets a well defined problem and uses appropriate
datasets. 

- Weaknesses:
The paper suffers from several drawbacks
1. The paper is hard to read due to incorrect usage of English. The current
manuscript would benefit a  lot from a review grammar and spellings. 
2. The main machine learning problem being addressed is poorly described. What
was a single instance of classification? It seems every transcripts was
classified as MCI or No MCI. If this is the case, the dataset descriptions
should describe the numbers at a transcript level. Tables 1,2, and 3 should
describe the data not the study that produced the transcripts. The age of the
patients is irrelevant for the classification task. A lot of text (2 pages) is
consumed in simply describing the datasets with details that do not affect the
end classification task. Also, I was unsure why numbers did not add up. For
e.g.: in section 4.1.1 the text says 326 people were involved. But the total
number of males and females in Table 1 are less than 100?
3. What is the motivation behind enriching the graph? Why not represent each
word by a node in the graph and connect them by the similarity between their
vectors, irrespective of co-occurrence?
4. The datsets are from a biomedical domain. No domain specific tools have been
leveraged.
5. Since dataset class distribution is unclear, it is unclear to determine if
accuracy is a good measure for evaluation. In either case, since it is a binary
classification task, F1 would have been a desirable metric. 
6. Results are reported unto 4 decimal places on very small datasets (43
transcripts) without statistical tests over increments. Therefore, it is
unclear if the gains are significant.

[Official Review · Reviewer 3 · rating 3 · confidence 3]
soundness 4 · originality 3 · clarity 3 · impact 2 · substance 4 · appropriateness 3 · meaningful comparison 3 · presentation format Poster

The paper describes a novel application of mostly existing representations,
features sets, and methods: namely, detecting Mild Cognitive Impairment (MCI) 
in speech narratives. The nature of the problem, datasets, and domain are
thoroughly described. While missing some detail, the proposed solution and
experiments sound reasonable. Overall, I found the study interesting and
informative.

In terms of drawbacks, the paper needs some considerable editing to improve
readability. Details on some key concepts appear to be missing. For example, 
details on the multi-view learning used are omitted; the set of “linguistic
features” needs to be clarified; it is not entirely clear what datasets were
used to generate the word embeddings (presumably the 3 datasets described in
the paper, which appear to be too small for that purpose…). It is also not
clear why disfluencies (filled pauses, false starts, repetitions, etc.) were
removed from the dataset. One might suggest that they are important features in
the context of MCI. It is also not clear why the most popular tf-idf weighting
scheme was not used for the BoW classifications. In addition, tests for
significance are not provided to substantiate the conclusions from the
experiments. Lastly, the related work is described a bit superficially. 

Detailed comments are provided below:

Abstract: The abstract needs to be shortened. See detailed notes below.

Lines 22,23 need rephrasing.            “However, MCI disfluencies produce
agrammatical speech impacting in parsing results” → impacting the parsing
results?

Lines 24,25: You mean correct grammatical errors in transcripts manually? It is
not clear why this should be performed, doesn’t the fact that grammatical
errors are present indicate MCI? … Only after reading the Introduction and
Related Work sections it becomes clear what you mean. Perhaps include some
examples of disfluencies.

Lines 29,30 need rephrasing: “as it a lightweight and language  independent
representation”

Lines 34-38 need rephrasing: it is not immediately clear which exactly are the
3 datasets. Maybe: “the other two: Cinderella and … “            

Line 70: “15% a year” → Not sure what exactly “per year” means…

Line 73 needs rephrasing.

Lines 115 - 117: It is not obvious why BoW will also have problems with
disfluencies, some explanation will be helpful.

Lines 147 - 149: What do you mean by “the best scenario”?

Line 157: “in public corpora of Dementia Bank” → a link or citation to
Dementia Bank will be helpful. 

Line 162: A link or citation describing the “Picnic picture of the Western
Aphasia Battery” will be helpful.

Line 170: An explanation as to what the WML subtest is will be helpful.

Line 172 is missing citations.

Lines 166 - 182: This appears to be the core of the related work and it is
described a bit superficially. For example, it will be helpful to know
precisely what methods were used to achieve these tasks and how they compare to
this study.

Line 185: Please refer to the conference citation guidelines. I believe they
are something along these lines: “Aluisio et al. (2016)  used…”

Line 188: The definition of “PLN” appears to be missing.

Lines 233 - 235 could you some rephrasing. Lemmatization is not necessarily a
last step in text pre-processing and normalization, in fact there are also
additional common normalization/preprocessing steps omitted. 

Lines 290-299: Did you create the word embeddings using the MCI datasets or
external datasets?

Line 322: consisted of → consist of

Lines 323: 332 need to be rewritten. ... “manually segmented of the
DementiaBank and Cinderella” →  What do you mean by segmented, segmented
into sentences? Why weren’t all datasets automatically segmented?; “ABCD”
is not defined; You itemized the datasets in i) and ii), but subsequently  you
refer to 3 dataset, which is a bit confusing. Maybe one could explicitly name
the datasets, as opposed to referring to them as “first”, “second”,
“third”.

Table 1 Caption: The demographic information is present, but there are no any
additional statistics of the dataset, as described.

Lines 375 - 388:  It is not clear why filled pauses, false starts, repetitions,
etc. were removed. One might suggest that they are important features in the
context of MCI ….

Line 399: … multidisciplinary team with psychiatrists ... → consisting of
psychiatrists…

Lines 340-440: A link or citation describing the transcription norms will be
helpful.

Section 4.2.1. It is not clear what dataset was used to generate the word
embeddings. 

Line 560. The shortest path as defined in feature 6?

Section “4.2.2 Linguistic Features” needs to be significantly expanded for
clarity. Also, please check the conference guidelines regarding additional
pages (“Supplementary Material”).

Line 620: “In this work term frequency was …” → “In this work, term
frequency was …” Also, why not tf-idf, as it seems to be the most common
weighting scheme? 

The sentence on lines 641-645 needs to be rewritten.

Line 662: What do you mean by “the threshold parameter”? The threshold for
the word embedding cosine distance?

Line 735 is missing a period.

Section 4.3 Classification Algorithms: Details on exactly what scheme of
multi-view learning was used are entirely omitted. Statistical significance of
result differences is not provided.